# Preclinical Models of Neuroendocrine Neoplasia

**DOI:** 10.3390/cancers14225646

**Published:** 2022-11-17

**Authors:** Andrew J. H. Sedlack, Kimia Saleh-Anaraki, Suresh Kumar, Po Hien Ear, Kate E. Lines, Nitin Roper, Karel Pacak, Emily Bergsland, Dawn E. Quelle, James R. Howe, Yves Pommier, Jaydira del Rivero

**Affiliations:** 1National Institute of Biomedical Imaging and Bioengineering, National Institutes of Health (NIH), Bethesda, MD 20892, USA; 2Medical Scientist Training Program, Feinberg School of Medicine, Northwestern University, Chicago, IL 60611, USA; 3National Cancer Institute, NIH, Bethesda, MD 20892, USA; 4Department of Surgery, University of Iowa, Iowa City, IA 52242, USA; 5Oxford Center for Diabetes, Endocrinology, and Metabolism, Radcliffe Department of Medicine, University of Oxford, Oxford OX3 7LE, UK; 6Eunice Kennedy Shriver National Institute of Child Health and Human Development, NIH, Bethesda, MD 20892, USA; 7University of California, San Francisco Helen Diller Family Comprehensive Cancer Center, San Francisco, CA 94158, USA; 8Department of Neuroscience and Pharmacology, University of Iowa, Iowa City, IA 52242, USA

**Keywords:** neuroendocrine tumor, genetically-engineered mouse model, organoid, cell line, small cell lung cancer, castration-resistant prostate cancer, xenograft

## Abstract

**Simple Summary:**

Neuroendocrine neoplasia comprise many distinct and rare subtypes of cancers. Preclinical models are essential for improving understanding of these diseases because clinical data is scarce. We review available preclinical models across a wide spectrum of neuroendocrine neoplasia (including those affecting the lungs, gastrointestinal system, prostate, and adrenal glands). We consider models of varying complexity and accuracy, covering both in vitro models such as cell lines and 3D models, and in vivo models such as xenografts and genetically-engineered mouse models. Better access and understanding of these models as provided in this work will help to enable research into pathology and treatment across the spectrum of neuroendocrine neoplasia.

**Abstract:**

Neuroendocrine neoplasia (NENs) are a complex and heterogeneous group of cancers that can arise from neuroendocrine tissues throughout the body and differentiate them from other tumors. Their low incidence and high diversity make many of them orphan conditions characterized by a low incidence and few dedicated clinical trials. Study of the molecular and genetic nature of these diseases is limited in comparison to more common cancers and more dependent on preclinical models, including both in vitro models (such as cell lines and 3D models) and in vivo models (such as patient derived xenografts (PDXs) and genetically-engineered mouse models (GEMMs)). While preclinical models do not fully recapitulate the nature of these cancers in patients, they are useful tools in investigation of the basic biology and early-stage investigation for evaluation of treatments for these cancers. We review available preclinical models for each type of NEN and discuss their history as well as their current use and translation.

## 1. Introduction

Neuroendocrine neoplasia (NENs) comprise a spectrum of malignant neoplasia that originate in neuroendocrine cells and can affect almost any part of the body. They vary from low Ki-67 level neuroendocrine tumors (NETs) to high Ki-67 level grade 3 NETs and biologically distinct neuroendocrine carcinomas (NECs). Although patients with low-grade tumors can survive many years, their high-grade counterparts are extremely aggressive and have a dismal outcome. As neuroendocrine cells are ubiquitous in our body, NENs can form in different organs including the gastrointestinal (GI) tract, pancreas, lungs, gallbladder, thymus, thyroid gland, testes, prostate, ovaries, and skin. NETs occur most often in the gastroenteropancreatic (GEP) system (51–67%) and lungs (21–27%) [1,2,3]. Gastroenteropancreatic NETs (GEP-NETs) occur most often in the small intestine (30–31%), rectum (26–29%), pancreas (12–13%), or appendix (6%) [2,3]. 

NENs can cause a broad range of symptoms based on type, location and secretion of hormones. They are classified as functional or non-functional NENs based on whether or not, respectively, they secrete hormones. Most functional NENs are NETs [1,4,5]. Rates of functionality differ depending on the type of NET: 10% of gastrointestinal, 10% of pulmonary, 10–30% of pancreatic [6,7], and 40% of adrenal [8]. While functional NETs may cause significant clinical signs and symptoms that often leads to earlier diagnosis, presentation varies widely depending on the hormone secreted. Carcinoid syndrome, which occurs in up to 20% of patients with small bowel (SB)NETs, is associated with flushing, abdominal pain, diarrhea, bronchoconstriction, and carcinoid heart disease [3]. In contrast, due to the lack of signs and symptoms, non-functional NETs are often diagnosed in the later stages after the occurrence of symptoms related to the mass effect of the tumor or metastases [9]. Moreover, while most NETs are sporadic, 20% are associated with hereditary genetic syndromes, such as multiple endocrine neoplasia type 1, von Hippel-Lindau, tuberous sclerosis, and neurofibromatosis type 1 [1]. 

NENs have been recognized for at least a century. They are considered an orphan disease (i.e., have a prevalence of <200,000 in the United States), and hence research is relatively sparse [10]. Disappointingly, survival of patients with NENs has not changed appreciably over the past three decades in either the USA or UK and without robust preclinical models, future drug development and our understanding of the underlying pathophysiology can be challenging. Indeed, an unmet need is improvement in the available models for each NEN subtype (i.e., both their number and authentic recapitulation of the human disease) that will enable rigorous investigation of their derivation, biology, behavior, and new treatment strategies.

For instance, small cell prostate carcinoma resembles neuroendocrine small cell lung cancer (SCLC) and makes up a minor fraction (<2%) of prostate cancers [11,12,13]. However, systemic therapies for prostate adenocarcinoma (the most commonly diagnosed type) are currently based on inhibition of androgen receptor signaling, which unfortunately leads to partial or complete neuroendocrine transdifferentiation in 10–20% of patients and accompanying development of resistance to hormonal treatment [13,14,15]. This highlights the biomedical relevance of studying neuroendocrine cell differentiation, but preclinical models of de novo small cell prostate carcinoma (natively neuroendocrine) are limited except for xenografts [16,17]. The comparatively high incidence of treatment-induced neuroendocrine prostate cancer (t-NEPC) makes up an increasing fraction of both prostate and neuroendocrine tumors in treatment (especially within neuroendocrine carcinomas but also of NENs overall) [13], warranting development of better models for investigation.

Disease models are indispensable in yielding meaningful insights into the etiology and biology of human neoplasms and to develop novel treatments. Herein, we consider the existing in vitro models of NENs including human-derived cell lines and 3D models (such as organoids, spheroids, and tumoroids), and in vivo NEN models including patient-derived xenografts (PDXs) and genetically engineered mouse models (GEMMs). 

While more models are needed, this review also summarizes how those available have nonetheless provided valuable insights into the pathogenesis and natural history of human NENs. Moving forward, it will be increasingly important that current and new NEN models be used to study methods of early tumor detection, molecular predictors of tumor behavior that can be leveraged clinically, and responses (including mechanisms) to novel targeted drug therapies.

## 2. Results

### 2.1. Cell Lines

Cell line propagation is the oldest and simplest method of tissue culture. Their convenience makes them one of the more accessible and flexible strategies for basic mechanistic research and drug screening. However, the huge gap between the microenvironment experienced by tumors in vivo and in monolayer or spheroid culture on polystyrene limits their efficacy as a model system. Less aggressive cancers tend to be more reliant on particular environmental cues and often prove difficult or impossible to culture as homogeneous cell lines, leading to overrepresentation of poorly differentiated carcinomas and other rapidly growing forms in cell line libraries [18]. Cell lines are presented in Table 1.

#### 2.1.1. NEN Cell Lines

Most NEN cell lines are derived from poorly differentiated carcinomas, whose aggressive behavior and rapid replication is better suited to long term in vitro culture than slow-growing well-differentiated NETs [66]. This unfortunately means that there is a lack of models for the latter types of tumors, although establishment of the first well-differentiated small bowel NET cell line (GOT1) and pancreatic NET cell line (NT-3) were reported [51,61]. Existing GEP-NET lines like CM, BON1, and KRJ-I exhibit significant karyotypic and genetic differences from most NETs, and may not be derived from enterochromaffin cells (although they may still have significant value as NEC models) [67,68,69,70]. 

Many NEN cell lines established in the late 20th century were derived from thyroid or lung primaries, and since then those developed have been from GEP-NETs. While development of new SCLC lines has continued [44], declining interest in such models reflects both decreasing smoking rates (and SCLC incidence) and increasing understanding of the limitations of 2D cell culture as a model system [35]. Increasingly, protocols for maintaining in vitro cell line models are established in parallel with PDX models of the same cells, a promising trend towards diversifying neuroendocrine research. Such combined models allow for assessment of multiple factors. For example, the first duodenal NEC cell line (TCC-NECT-2, 2018) induced cachexia in a PDX mouse model by an unknown mechanism, which invites further study that may have been missed if it had only been used in vitro [64,71]. Increasing accessibility of 3D culture techniques such as spheroids or organoids may also contribute to improved ability to recapitulate NEN characteristics. The recently established SS-2 line showed greater expression of cancer stem cell markers in spheroids than in adherent conditions [28]. Debate over the validity of data based on cell lines affirms their role as part of a complementary set of different models, which are critical to piecing together the molecular mechanisms of NEN, but best used in combination with other models [18].

While many lines have been derived and immortalized from NEN, few of them continue to behave like NEN in standard tissue culture. Many cell lines derived from functional NENs rapidly lose secretory function in vitro [48]. Some existing cell lines have been discovered upon closer examination to actually be Epstein-Barr virus-transformed lymphoblastoid cells (L-STS, H-STS, KRJ-I) [72]. However, it remains unclear what qualities are best used to describe NEN cells—between genetics, function, and RNA-seq identified behavior [73]. For example, transcriptomic and secretomic comparison of BON1 and KRJ-I suggests that the latter line is phenotypically more neuroendocrine, while immunohistochemical comparison suggests the latter is actually of lymphoblastoid origin [70,72]. Contradictory results like these show that while the development of new cell lines and models for neuroendocrine research is exciting, it is also important to better investigate the validity of existing models and the research based off them.

#### 2.1.2. NEPC Cell Lines

The rising prevalence of neuroendocrine prostate cancer (NEPC) is generally thought to be due to use of androgen receptor (AR) signaling inhibitors in treatment of prostate adenocarcinoma [14]. In vitro, many methods have been established for reliable transdifferentiation from adenocarcinoma into neuroendocrine phenotypes [74,75,76]. This review does not go into great detail about prostate adenocarcinoma cell lines which may be transdifferentiated as opposed to established NEPC lines.

#### 2.1.3. PPGL Cell Lines

Pheochromocytomas (PCC, referred to in combination with paragangliomas as PPGL) have historically proven challenging to establish as cell lines. The first human pheochromocytoma cell lines, KNA and KAT45, were reported in 1998 [19,20]. No further pheochromocytoma models were developed until 2013, when a slow-growing, benign tumor was successfully immortalized via lentiviral infection with human telomerase reverse transcriptase, producing the hPheo1 cell line [21]. While this method may affect the behavior of the underlying cells, it shows potential for allowing research into the mechanisms of such slow-growing tumors that typically struggle in vitro, and has been reproduced with other pheochromocytomas [77]. Murine models of PPGL such as MTT and MPC have also proven valuable since human cell lines are so sparse [78]. 

### 2.2. 3D Models

Organoid, tumoroid, and other 3D tissue culture strategies recapitulate the tumor microenvironment to a greater degree than ordinary cell culture at the cost of added expense and complexity per sample. Increasingly, automation and standardized, well-characterized protocols for organoid establishment have allowed a greater degree of tumor types to be captured in organoid lines and used for research [79]. While traditional cell lines remain the simplest and most commonly used model, organoids are a valuable step up that can help to characterize especially more indolent tumor types that struggle to grow in conventional tissue culture. Organoid systems are presented in Table 2. 

#### 2.2.1. NEN 3D Models

Individually developed 3D models are valuable for understanding which parameters affect samples’ ability to survive as organoids. Recent models in particular have focused on directly transitioning patient tissue into 3D growth environments to better recapitulate the tumor sources [87]. Library and biobank projects that standardize collection and seeding procedures across multiple centers have also contributed greatly to the number of different models available [79]. In particular, Fuji et al.’s recent work comparing the growth of each tissue sample across a matrix of different media supplements provides greater context to the different possible neuroendocrine niches [80]. As more NEN organoid models are developed, the heterogeneity and different growth factors needed by different tumors types will be further elucidated—in particular, currently the vast majority of NET (as opposed to NEC) cells fail in vitro, and developing models which provide them a suitable niche for growth should be a priority in the near future [80,89]. Recent advances have also been made in generation of small cell lung cancer (SCLC) organoids and a single lung-derived non-small cell neuroendocrine carcinoma [81,82].

#### 2.2.2. NEPC 3D Models

Neuroendocrine prostate cancer organoids were first reported in 2018, in 3D culture using Matrigel [84]. The same organoids were also reimplanted as patient-derived organoid xenografts (PDOXs) [85]. Projects spanning in vitro and in vivo techniques like this by using PDOXs in conjunction with the original organoids help to find more reliable results from multiple convergent types of evidence. Follow-up studies have developed a defined medium for use in 3D culture based on functionalized polyethylene glycol hydrogel [86]. Defined media presents an opportunity to better understand the different niches in which different subtypes of tumor may flourish. Data transparency is particularly helpful in such projects—reporting not only which defined media mixtures worked but also all options which were tested [79,80]. 

PDOXs and seeding of PDX tissue into organoid culture allows a certain degree of interchange between these types of models, although it is unclear to what extent characteristics of the tumors are preserved during such changes. One advantage that organoids have over PDXs is the use of automation and robotics to achieve high throughput. One recent study derived organoids from the MURAL prostate cancer PDX biobank to compare many organoids in parallel, at scales which would be impossible in vivo [90].

#### 2.2.3. PPGL 3D Models

There are no published manuscripts on PPGL 3D models [91,92,93]. 

### 2.3. Patient-Derived Xenografts

Many of the previously discussed models have been used successfully in xenografts after some in vitro passaging (which has been discussed elsewhere in detail for GEP-NENS), but xenografts of samples acquired directly from biopsy or surgery into immunodeficient mice is typically seen as the gold standard of comparative evidence [18,94]. Patient derived xenografts are presented in Table 3. 

#### 2.3.1. NEN PDXs

Although the first NEN PDX models (TEG13 and TSG13) were established in 1995, there was relatively little development in this area for the following two decades [96]. Compared to prostate cancers, where biobanks increasingly standardize protocols and collection procedures across regions, new NEN PDXs have been developed incrementally and by disparate methods [99,100,102,116,117]. Recently, 2 GEP-NEC PDX models have been described by Tran et al. [97]. Beyond drug screening, some NEN PDXs have also been used to analyze circulating cancer cells [116]. Since zebrafish mature more rapidly than mice but conserve many aspects of mammalian vasculature, there has recently been increased interest in their use as a model for angiogenesis in NEN development and invasion [99]. One organoid NEN biobank has also developed companion PDXs for their models, covered in more detail in that section [79]. To date, no studies have successfully establishd well-differentiated NENs in PDX lines.

#### 2.3.2. NEPC PDXs

Historically, limited models for neuroendocrine prostate cancer (NEPC) specifically have been developed. With the advent of biobanks operating on standardized protocols, increasing numbers of NEPC models have become available [111,112,113]. Many prostate adenocarcinomas have been observed to transdifferentiate to a neuroendocrine phenotype under androgen starvation [111]. Of particular interest, Faugeroux et al. showed that circulating tumor cell explants from non-NE primaries may develop into NE tumors [55].

#### 2.3.3. PPGL PDXs

One successful set of PPGL PDXs has been generated, although via a technically complex protocol [103]. Recent progress also includes a high-fidelity *SDHb^−^*^/*−*^ rat-to-mouse xenograft [93].

### 2.4. Genetically-Engineered Mouse Models

Neuroendocrine tumors are some of the most frequently inherited types of cancer, and are associated with more than ten different genetic syndromes [118]. These range from general oncogene/tumor suppressor genes like *TP53* or *RB1* to ones associated with systemic neuroendocrine disease like *MEN1*, *VHL*, or *NF1* to highly specific risk factors like *SDHx* and *EPAS1*.

While cell lines and organoid models can be highly valuable in studying treatment and improving understanding of disease states, genetically engineered mouse models provide a window into tumorigenesis itself, in particular from a genetic perspective [119]. Early models often focused on systemic knockouts or transgenes, but engineered systems with such features as Cre/lox recombinases or drug-inducible promoters allow for precise spatial and temporal control of models [120]. These tools are especially useful with regard to genes that have large systemic effects like *TP53*, allowing researchers to isolate changes to it to a particular location or organ [121]. Similarly, while mouse models of heritable neuroendocrine disorders are valuable in recapitulating the overall progression of the disease, more specific models help to refine understanding of highly complex disorders [121,122]. *MEN1* mutations are most often associated with pancreatic, parathyroid, and pituitary tumors [123]. MEN2-associated (*RET*) mutations are most often associated with medullary thyroid carcinoma, pheochromocytoma, and parathyroid tumors [124]. *NF1* mutations are most often associated with adrenal, gastroenteropancreatic, and parathyroid tumors [125].

In the following section, we focus specifically on models engineered in mice. Notable genetic NEN models have also been developed in other organisms, in particular zebrafish. These are not discussed below for brevity but are referenced here for completeness [126,127,128,129,130,131,132,133,134,135,136].

#### 2.4.1. NEN GEMMS

Specific models of NEN subtypes are lacking since many NENs are interrelated or poorly understood in terms of molecular drivers [122,137]. Use of organ-specific Cre/lox and drug-induced systems has been critical in developing more specific NEN models, although most today still focus on pancreatic NETs [120]. Many models also link SV40-Tag oncogenes to particular promoters to facilitate local expression. For pancreatic NETs the RIP-Tag paradigm and its successors have been especially well-studied [138,139,140]. Gastric NET models have been developed that induce tumorigenesis through both overexpression and inhibition of gastrin-related genes [141,142]. There is a distinct lack of specific GEMMs for intestinal NETs (even though they are among the most prevalent GEP-NETs) [122]. Current models are based on SV40-Tag driven by glucagon or intestinal trefoil factor promoters [143,144]. As lung tumors are rarely resected, GEMMs are an essential tool in lung NEN research [122,145]. Many current lung NEC models focus on selective knockouts of *TP53*, *RB1*, or *PTEN* to recapitulate SCLC or LCNEC [146,147]. Genetically engineered mouse models are presented in Table 4. 

#### 2.4.2. NEPC GEMMS

Before treatment-induced neuroendocrine transdifferentiation of prostate cancer was well understood, GEMMs that modeled transdifferentiation of prostate adencarcinomas to NE phenotypes were seen as flawed, even though this has been since been observed clinically [240,273]. As t-NEPC has become more common in recent years, there has been renewed interest in models that feature this neuroendocrine transdifferentiation [274,275]. Most models depend on oncogenes driven by variations of probasin or PSP94 promoters [232,236,239,241,276].

#### 2.4.3. PPGL GEMMS

While models for genes of cluster 2 PPGL (*RET*, *NF1*, and *TMEM127*) have been studied since 1992, these models are suboptimal as they are generally associated with development of additional unrelated non-neuroendocrine tumors [92]. Historical attempts at generating GEMMS of cluster 1 pseudohypoxia (*VHL*, *SDH*) PPGL have failed to develop tumors or are embryonically lethal [277]. However, recent developments have changed this: The first successful pseudohypoxic PPGL GEMM was recently developed with a gain-of-function *EPAS1* mutation, characterizing Pacak-Zhuang syndrome [148], and tetracycline-induced dual *SDHB/NF1* mouse model was recently developed (showing that *SDHB* inactivation alone was insufficient for tumorigenesis, but coupling with *NF1* lead to pheochromocytoma) [91,151]. 

## 3. Conclusions

NENs constitute a heterogeneous and complex set of diseases. This feature demands a similarly heterogenous set of models to span the genetic and phenotypic diversity of the diseases. Since many forms are quite low in incidence, patient-derived models and clinical data about these diseases is at a premium, and preclinical models are all the more necessary to develop treatments for these patients with distinct but still lethal disease. Since aggressive disease is often better suited to in vitro growth and allows for more rapid assessment in mouse models, models are lacking for more indolent forms of neuroendocrine neoplasia, among all subtypes, but especially PPGL. 

In this review, we have summarized available preclinical models for NENs, their history, as well as current use and translation. While no single model can fully recapitulate disease in the human body, convergent evidence from multiple different types of models brings us closer to that understanding. In spite of the many models developed, there is relatively little sharing and combination of those models for such integrated investigations. Future work would benefit from horizontal use of multiple types of models by researchers via increased collaborative sharing of data and models, including difficulties encountered during their generation. Large biobank projects are helpful in reaching a wide variety of a single type of model, but current biobanks have not made efforts to capture heterogeneity between model types as well as between individual models. Cell lines are perhaps the easiest models to work with for more aggressive disease types that grow well in simple in vitro culture, however, they may not reflect the actual phenotype of disease in vivo, and many patient-derived cells fail to thrive in continuous culture. Organoid or tumoroid models that better recapitulate the tumor microenvironment can allow for growth of a greater variety of tumor types in vitro, and better model their biology and responses to treatment. Immunodeficient PDX models further capture interactions between tumors and other tissues and in particular routes of metastasis. While immunocompetent mouse models are more complex and less often successful, they further allow for modeling tumor-immune interactions which are increasingly critical in understanding the potential of modern immunotherapy. Finally, genetically engineered mouse models fill a critical niche in understanding the pathogenesis of tumors, in particular in combination with spatial and temporal controls to introduce related mutations in a controlled fashion within mouse tissue. 

Since neuroendocrine neoplasia represent such a genetically and phenotypically diverse group of pathologies, a diverse selection of models is also necessary to improve our understanding and treatment of them. This review underscores the conclusion that, despite persistent efforts by many groups, the establishment and application of NEN disease models have been limited. One reason is the indolent nature of these slowly growing tumors that make them difficult to propagate in culture or in animals, but another key factor is the rare nature of this disease and relatively small number of patients. Considering the recent rise in incidence (a 6.4-fold increase between 1973 and 2012 [2]), it is important to develop NEN disease models that more accurately reflect the biology of human NEN tissues in terms of diagnostic criteria and genetic alterations [89]. Much of the rise in incidence is thought to be the result of improved detection [2]. A greater diversity of model types and origins will help to address our understanding of these diseases in context of their increasing prevalence. All types of model systems are valuable at different parts of the drug screening and development process. Use and continued improvement of these models will drive preclinical advancements across the spectrum of neuroendocrine neoplasia.

## Figures and Tables

**Table 1 cancers-14-05646-t001:** Human neuroendocrine neoplasm-derived cell lines.

Cell Line	Source ^1^	Type	# Refs	Year	References
KNA	AM	PCC	3	1998	[19]
KAT45	AM	benign	-	1998	[20]
hPheo1	AM	benign	-	2013	[21]
NEC-DUE3	An	small cell carcinoma LN met	1	2018	[22]
NEC913, NEC1452	AP, C	LN met	-	2019	[23]
COLO320	C	colorectal w NE features	18	1979	[24]
LCC-18	C	colon	1	1991	[25]
NEC-DUE2	C	lymph node met	1	2014	[26]
HROC57	C	PD large cell carcinoma	1	2018	[27]
SS-2	C	ascending colon carcinoma	1	2019	[28]
EPG1	CB	-	-	1992	[29]
ECC18	E	esophageal	1	1993	[30]
TYUC-1	E	small cell carcinoma	3	2015	[31]
NEC-DUE1	GEJ	carcinoid hepatic met	1	2014	[26]
PTJ64p	JT	benign	-	2013	[32]
OAT	L	SCLC	-	1971	[33]
SHP-77	L	SCLC	17	1978	[34]
DMS-273	L	SCLC	18	1978	[35]
COR-L24	L	SCLC	7	1985	[36]
COR-L47	L	SCLC	9	1985	[36]
COR-L51	L	SCLC	7	1985	[36]
SCLC-21H	L	SCLC	13	1987	[37]
CPC-N	L	SCLC	10	1992	[38]
UMC-11	L	carcinoid	9	1992	[39]
NCI-H720	L	atypical carcinoid	13	1992	[39,40]
NCI-H727	L	bronchial carcinoid	23	1992	[39]
COR-L103	L	SCLC	3	1992	[41]
COR-L266	L	SCLC	1	1992	[41]
COR-L279	L	SCLC	12	1992	[41]
NCI-H82	L	SCLC	40	1996	[42]
NCI-H0446	L	SCLC	33	1996	[42]
NCI-H0510	L	SCLC	23	1996	[42]
NCI-H0524	L	SCLC	25	1996	[42]
NCI-H1105	L	SCLC	10	1996	[42]
NCI-H1436	L	SCLC	12	1996	[42]
NCI-H1694	L	SCLC	12	1996	[42]
NCI-H1930	L	SCLC	12	1996	[42]
NCI-H1963	L	SCLC	17	1996	[42]
NCI-H2029	L	SCLC	11	1996	[42]
NCI-H2171	L	SCLC	26	1996	[42]
NCI-H2196	L	SCLC	10	1996	[42]
HCC33	L	SCLC	15	1998	[43]
SCLC-J1	L	SCLC	-	2021	[44]
QGP1	Pa	delta-islet carcinoma	22	1980	[45]
CM	Pa	insulinoma	5	1987	[46]
BON1	Pa	carcinoid LN met	12	1991	[47]
HuNET	Pa	VIP-secreting	2	2001	[48]
A99	Pa	small cell carcinoma	3	2011	[49]
APL1	Pa	pancreatic liver met	-	2016	[50]
NT-3	Pa	WD carcinoid LN met	2	2018	[51]
NT-18P	Pa	NET	-	2022	[52]
NT-18LM	Pa	liver met NT-18P	-	2022	[52]
NT-36	Pa	recurrence of NT-18P	-	2022	[52]
NT-32	Pa	pancreatic NEC	-	2022	[52]
LnCaP	Pr	-	-	1980	[53]
NCI-H660	Pr	small cell prostate cancer	15	1989	[54]
Faugeroux 2020	Pr	treatment-induced	-	2020	[55]
KUCaP13	Pr	treatment-induced	-	2021	[56]
N-TAK-1	R	rectal carcinoma	-	1999	[57]
NECS-P	R	rectal carcinoma	1	2000	[58]
NECS-L	R	rectal carcinoma liver met	1	2000	[58]
KRJ-I	SB	ileal carcinoid	8	1996	[59]
CT-nu-1	SB	atypical duodenal carcinoid	-	1998	[60]
GOT1	SB	ileal carcinoid liver met	3	2001	[61]
CNDT2	SB	midgut carcinoid liver met	3	2007	[62]
P-STS	SB	ileal carcinoid primary	4	2009	[63]
L-STS	SB	ileal carcinoid LN met	4	2009	[63]
H-STS	SB	ileal carcinoid liver met	3	2009	[63]
TCC-NECT-2	SB	duodenal carcinoma	2	2018	[64]
NT-38	SB	duodenal NEC	-	2022	[52]
MTC-F	T	MTC	2	1990	[65]
MTC-SK	T	MTC	6	1990	[65]

^1^ Adrenal medulla (AM), anus (An), Ampullary (AP), colon (C), carotid body (CB), esophagus (E), gastroesophageal junction (GEJ), jugulotympanic (JT), lung (L), pancreas (Pa), pituitary (Pi), prostate (Pr), rectum (R), small bowel (SB), stomach (St), thyroid (T), unknown (U).

**Table 2 cancers-14-05646-t002:** Human neuroendocrine neoplasm-derived organoids.

PDX Line	Host	Source ^1^	Successes ^2^	Attempts ^2^	Year	References
NEC913, NEC1452	spheroid	AP, C	1	1	2019	[23]
CRC14	organoid	C	1	1	2016	[80]
Kawasaki 2020	organoid	GEP-NEC	16	23	2020	[79]
Kawasaki 2020	organoid	GEP-NET	3	16	2020	[79]
Kim 2019	organoid	L	3	3	2019	[81]
Gmeiner 2020	organoid	L	4	4	2020	[82]
April-Monn 2021	organoid	Pa	6	7	2021	[66]
MSK-PCa4	organoid	Pr	7	32	2014	[83]
OWCM	organoid	Pr	4	25	2018	[84,85,86]
CRC19	organoid	R	1	1	2016	[80]
Dijkstra 2021	organoid	St/C	1	3	2021	[87]
ANI-27S	spheroid	U	1	1	2017	[88]

^1^ Adrenal medulla (AM), anus (An), Ampullary (AP), colon (C), carotid body (CB), esophagus (E), gastroesophageal junction (GEJ), jugulotympanic (JT), lung (L), pancreas (Pa), pituitary (Pi), prostate (Pr), rectum (R), small bowel (SB), stomach (St), thyroid (T), unknown (U). ^2^ “Successes” denotes models established for serial passaging, and “attempts” denotes number of tumor explants which failed to survive (to the extent reported on by the primary article in question).

**Table 3 cancers-14-05646-t003:** Neuroendocrine neoplasm patient-derived xenografts.

PDX Line	Host	Source ^1^	Successes ^2^	Attempts ^2^	Year	References
SJ-ACC3	CB17 *scid*^−/−^	AC	1	1	2013	[95]
HROC57	NMRI nu/nu mice	C	1	1	2018	[27]
TEG13	athymic nude mice	E	1	1	1995	[96]
Kawasaki 2020	NOG mice	GEP-NEC	15	22	2020	[79]
Tran 2022	NSG mice	GEP-NEC	2	6	2022	[97]
Yang 2016	NOD/SCID mice	GEP-NET	106	6	2016	[98]
Gaudenzi 2017	Tg(fli1a:EGFP) y1 zebrafish	GEP-NET	2	3	2017	[99]
Anderson 2015	NOD/SCID mice	L	8	12	2015	[100]
LXFS	NOG Taconic mice	L	1	1	2021	[101]
HNV PDX-PNET	athymic nude mice	Pa	1	1	2018	[102]
Gaudenzi 2017	Tg(fli1a:EGFP) y1 zebrafish	Pi	1	6	2017	[99]
Powers 2017	NSG mice	PPGL	3	13	2017	[103]
UCRU-PR-2	nude mice	Pr	1	1	1987	[104,105,106]
WISH-PC2	SCID mice	Pr	1	1	2000	[107,108]
WM-4A	SCID mice	Pr	1	1	2008	[108]
MDA PCa	CB17 SCID mice	Pr	5	11	2011	[16,109,110]
KUCaP13	SCID mice	Pr	1	1	2014	[56]
LTL	NOD/SCID mice	Pr	7	18	2014	[111]
LuCaP	Nu/Nu or CB17 SCID mice	Pr	4	^2^	2017	[112]
MURAL	NSG or NOD/SCID mice	Pr	30	^2^	2018	[113,114]
Faugeroux 2020	NSG mice	Pr	7	15	2020	[55]
EN-1	nude mice	SB	1	1	1998	[60]
TSG15	athymic nude mice	St	1	1	1995	[96]
HuPrime GA	Balb/c nude mice	St	20	^2^	2013	[115,116]

^1^ Adrenal medulla (AM), anus (An), colon (C), carotid body (CB), esophagus (E), gallbladder (G), gastroesophageal junction (GEJ), jugulotympanic (JT), lung (L), pancreas (Pa), pituitary (Pi), prostate (Pr), rectum (R), small bowel (SB), stomach (St), thyroid (T), unknown (U). ^2^ Fraction of attempts which were NE not specified. ^2^ “Successes” denotes models established for serial passaging, and “attempts” denotes number of tumor explants which failed to survive (to the extent reported on by the primary article in question).

**Table 4 cancers-14-05646-t004:** Genetically-engineered mouse models of neuroendocrine neoplasia.

Model ^1^	Organ ^2^	Neoplasia Type	Model Type ^3^	Gene (Promoter ^4^)	Year	References
EPAS1	-	polyhormonal, polycythemia	transgenic	*EPAS1^A529V^*	2019	[148]
Nf1^+/−^	AM	PCC, leukemia	heterozygous KO	*NF1*	1994	[149]
NF1^+/−^	AM	pheo	heterozygous KO	*NF1*	2016	[92]
NES-VHL	AM	PGL	TS inducible KO	*VHL* (*NES*)	2017	[150]
SDHB^f/f^NF1^f/f^Rosa^mt/mg^/+Th-Cre	AM	PCC	multiple KO	*SDHB*, *NF1*	2021	[151]
Ink4a Arf^+/+^ Pten^+/−^ Ink4a Arf^+/−^ Pten^+/−^ Ink4a Arf^−/−^ Pten^+/−^	AM, C, L	PCC, NEC, NEPC	multiple KO	*CDKN2A*, *ARF*, *PTEN*	2002	[152]
RET-KO	AM, T	PCC, MTC	KI	*RET*	2000	[153]
ITF-Tag	C	NEC	transgenic	*SV40-Tag* (*ITF*)	2004	[144]
CC10-hASH1	L	SCLC, NSCLC	transgenic	*ASCL1* (*SCGB1A1*)	2000	[154]
RB-TP53-KO	L	SCLC, LCNEC	homozygous KO	*RB1*, *TP53*	2003	[155]
RB-TP53-RB1-KO	L	SCLC, LCNEC	homozygous KO	*RB1*, *TP53, RB1*	2010	[156]
RP-TP53-PTEN-KO	L	SCLC, LCNEC, NSCLC	multiple KO	*RB1*, *TP53*, *PTEN*	2014	[146]
RIP-Tag	Pa	NET	transgenic	*SV40-Tag* (*RIP2*)	1985	[157]
RIP-Tag2 (Tg(RIP1-Tag)2Dh)	Pa	NEN various	transgenic	*SV40-Tag* (*RIP*)	1985	[143,157,158]
VT-C (Avp-Tag)	Pa	insulinoma	transgenic	*SV40-Tag* (*AVP*)	1987	[159]
SV-202	Pa	insulinoma	transgenic	*SV40-Tag* (*MT*)	1989	[160]
ELSV (Tg(Ela-l, SV4OE)Bril8)	Pa	insulinoma, D cell hyperplasia	transgenic	*SV40-Tag* (*EL*)	1990	[161]
L-PK/Tag (Tg(Pklr-Tag)Ak)	Pa	islet cell carcinoma	transgenic	*SV40-Tag* (L-type pyruvate kinase)	1992	[162]
GP1.5 Tag, GP10.5 Tag	Pa	insulinoma, ductal hyperplasia	transgenic	*SV40-Tag* (*GAST*)	1993	[163]
RIP-Tag5 (Tg(RIP1-Tag)5Dh)	Pa	insulinoma/invasive carcinoma	transgenic	*SV40-Tag* (*RIP*)	1996	[164]
RIP-MyrAkt1	Pa	NET	transgenic	*MyrAKt1* (*RIP*)	2001	[165,166]
Cdk4R24C/R24C (Cdk4tm1.1Bbd/Cdk4tm1.1Bbd)	Pa	insulinoma	homozygous KI	*CDK4* ^R24C^	2001	[167]
pIns-c-MycERTAM/RIP-Bcl-xL	Pa	islet cell carcinoma	TS transgenic	*MYC, BCL-xl* (*RIP*)	2002	[168]
Men1^T/T^; Men1^T/+^ (Men1^tm1Zqw^/Men1+)	Pa	polyhormonal	heterozygous KO	*MEN1*	2003	[169]
Men1^F/F^-RipCre+ (Men1^tm1.2Zqw^/Men1^tm1.2Zqw^ Tg(Ins2-cre)23Herr/0)	Pa	islet cell carcinoma	homozygous KO	*MEN1* (*RIP*)	2003	[169,170]
Gcgr^−/−^	Pa	glucagonoma, exocrine hyperplasia	homozygous KO	*GCGR*	2003	[171]
elastase-tv-a; RCAS-c-myc; p16/p19^−/−^	Pa	NET	transgenic	*MYC*, *INK4a/ARF* (*EL*)	2003	[172]
elastase-tv-a; RCAS-PyMT; p16/p19^−/−^	Pa	progenitor	transgenic	*pyMT*, *INK4a/ARF* (*EL*)	2003	[172]
Prdx1^−/−^ (Prdx1^tm1Rave^/Prdx1^tm1Rave^)	Pa	adenoma	homozygous KO	*PRDX1*	2003	[173]
Men1^loxP/loxP^ Rip-cre+ (Men1^tm1Gfk^/Men1^tm1Gfk^ Tg(Ins2cre)25Mgn)	Pa	insulinoma	TS homozygous KO	*MEN1* (*RIP*)	2004	[174]
Men1^+/−^; Rb1^ΔX2/+^ (Men1^tm1.1Gfk^ /Men1+Rb1^tm1Tyj^ /Rb1+)	Pa	insulinoma	heterozygous KO	*MEN1*	2007	[175]
RIP-MyrAkt1 (Tg(Ins2-Akt1 *)3Mbb)	Pa	insulinoma, islet cell carcinoma	transgenic	*AKT1* (*RIP*)	2008	[166]
Men1^tm1Rvt^/Men1+	Pa	insulinoma	heterozygous KO	*MEN1*	2009	[176,177]
Pdx1-Cre; Men1^f/f^ (Men1^tm1Ctre^/Men1^tm1Ctre^; Tg(Pdx1-cre)89.1Dam/0)	Pa	insulinoma	homozygous KO	*MEN1* (*PDX1*)	2009	[178]
Pdx1-Cre; Vhl^f/f^ (Vhl^tm1Lss^/Vhl^tm1Lss^; Tg(Pdx1-cre)89.1Dam/0)	Pa	adenoma	TS homozygous KO	*VHL* (*PDX1*)	2009	[179]
Men1^F/F^-GluCre+	Pa	mixed	TS homozygous KO	*MEN1* (*RG*)	2010	[180]
Glu-Cre;Men1^f/f^	Pa	insulinoma	TS homozygous KO	*MEN1* (*RG*)	2010	[181]
RipTag-IRES-Luciferase (RTL1 ) (Tg(Ins1-Tag, -luc)1Gcr)	Pa	insulinoma/invasive carcinoma	transgenic	*SV40-Tag* (*RIP*)	2010	[138]
RIP-Tag (Tg(RIP1-Tag)2Dh)	Pa	insulinoma/invasive carcinoma	transgenic	*SV40-Tag* (*RIP*)	2010	[140]
Gcgr^−/−^ (Gcgr^tm1Jcp^/Gcgr^tm1Jcp^)	Pa	glucagonoma	homozygous KO	*GCGR*	2011	[182]
Cul9^tm1.2Yxi^/Cul9+	Pa	insulinoma	heterozygous KO	*CUL9*	2011	[183]
PDX1-MEN	Pa	NET	TS homozygous KO	*MEN1* (*PDX1*)	2012	[184]
RIP-TβAg (Tg(Ins2-Tag*, FLPe)#Gne)	Pa	insulinoma/invasive carcinoma	transgenic	*SV40-Tag* (*RIP*)	2012	[185]
Pc2^−/−^ (Pcsk2^tm1Dfs^/Pcsk2^tm1Dfs^)	Pa	adenoma	homozygous KO	*PCSK2*	2014	[186]
RenCre; Tp53^loxP/loxP^ Rb^loxP/loxP^	Pa	glucagonoma, unrelated sarcoma	TS homozygous KO	*TP53*, *RB1* (*REN*)	2014	[187]
Gcg^gfp/gfp^	Pa	islet cell carcinoma	multiple KO	*GCG*	2015	[188]
Fabpl-Cre^tg/+^Rb^c/c^ (Tg(Fabp1-cre)1Jig/Rb^fl/fl^)	Pa	NEC	TS homozygous KO	*RB1* (*FABP1*)	2015	[189]
RIP7-rtTA; tet-o-MT; p48-Cre; p16/p19^loxP/loxP^	Pa	NET	transgenic	*PyMT*, *INK4A/ARF* (*RIP*, *PTF1A*)	2016	[190]
RIP7-rtTA; tet-o-MT; p48-Cre;p53^loxP/loxP^	Pa	NET	transgenic	*PyMT*, *TP53* (*RIP*, *PTF1A*)	2016	[190]
Pdx1-tTA; tet-o-MT; p48- Cre; p16/p19^lox/lox^	Pa	NET	transgenic	*PyMT*, *INK4A/ARF* (*PDX1*, *PTF1A*)	2016	[190]
Pdx1-tTA; tet-o-MT; p48-Cre;p53^lox/lox^	Pa	mixed acinar cell carcinoma/NE	transgenic	*PyMT*, *p53* (*PDX1*, *PTF1A*)	2016	[190]
Men1^L/L^/RIP2-CreER (Tg(Ins2-cre/ERT)1Dam/J, (RIP2-CreER), Men1^tm1.1Ctre^/J)	Pa	insulinoma	TS inducible KO	*MEN1* (*RIP2*)	2017	[120]
RIP-Tag (Tg(RIP1-Tag)2Dh)	Pa	pNET nonfunctional	transgenic	*SV40-Tag* (*RIP*)	2019	[139]
Men1^flox/flox^ Pten^flox/flox^ RIP-Cre(Men1^tm1.2Ctre^, Pten^tm1Hwu^; Ins1^tm1.1^(cre)Thor)	Pa	insulinoma	TS homozygous KO	*MEN1, PTEN* (*RIP*)	2020	[191]
Men1^flox/flox^ Pten^flox/flox^ MIP-Cre(Men1^tm1.2Ctre^, Pten^tm1Hwu^; Ins1^tm1.1^(cre)Thor)	Pa	insulinoma	TS homozygous KO	*MEN1*, *PTEN* (*MIP*)	2020	[191]
Pdx1-Cre; Rb^f/f^	Pa	NET	TS homozygous KO	*RB1 (PDX1)*	2020	[192]
Pdx1-Cre; Trp53^R172H^; Rb^f/f^	Pa	NET	TS homozygous KO	*TP53*, *RB1 (PDX1)*	2020	[192]
INS-p25OE	Pa	NET	TS inducible KI	*CDK5R1 (IN2/tetOp)*	2021	[193]
Glu2-Tag	Pa, C	NET	transgenic	*SV40-Tag* (*RG*)	1988	[194]
RIP-Tag/RIPPyST1	Pa, C	NET	transgenic	*SV40-Tag* (*RIPPyST1*)	1991	[143]
GLUTag-Y Tg(Gcg-TAg)25Ddr	Pa, C	invasive carcinoma, glucagonoma	transgenic	*SV40-Tag* (*RG*)	1992	[195,196]
TP53^+/−^ RB^+/−^/TP53^−/−^RB^+/−^	Pa, Pi, T	MTC, ICC, lymphoma	multiple KO	*TP53*, *RB1*	1994	[197]
MEN1^+/−^	Pa, Pi, T, PT, AM	NET	heterozygous KO	*MEN1*	2011	[198]
RIP-Tag/RIPPyST	Pa, SB	polyhormonal invasive carcinoma	transgenic	*SV40-Tag*, *PyST1* (*RIP*)	1990	[158]
RIP-Tag (Tg(RIP1-Tag)2Dh)	Pa, SB	pNET and siNET	transgenic	*SV40-Tag* (*RIP*)	2020	[199]
Secretin-Tag	Pa, SB, C	polyhormonal NEN	transgenic	*SV40-Tag* (*SCT*)	1995	[200]
GHRH-MT	Pi	polyhormonal	transgenic	*GHRH* (*MT*)	1992	[201]
AVP/SV40	Pi	polyhormonal	transgenic	*SV40* (*AVP*)	1992	[202]
CRH-MT	Pi	corticotropinoma	transgenic	*CRH* (*MT*)	1992	[203]
PyLT-1	Pi	corticotropinoma	transgenic	*PyLT* (*Py* early region)	1992	[204]
POMC-SV40	Pi	corticotropinoma	transgenic	*PyLT-SV40* (*POMC*)	1993	[205]
Cdkn1b^+/-^	Pi	somatotropinoma	heterozygous KO	*CDKN1B*	1996	[206,207,208]
p18INK4c	Pi	somatotropinoma	heterozygous KO	*CDKN2C*	1998	[209]
p18^−/−^	Pi	corticotropinoma	homozygous KO	*CDKN2C*	1998	[209]
p18^−/−^p27^−/−^	Pi	corticotropinoma	homozygous KO	*CDKN2C*, *CDKN1B*	1998	[209]
hFSHB-SV40tsTag	Pi	gonadotropinoma non-functioning adenomas	transgenic	*SV40-Tag* (*FSHb*)	1998	[210]
Rb^−/−^	Pi	-	TS KO	*RB1*	1998	[211,212]
Men1^TSM/+^	Pi	prolactinoma	multiple KO	*MEN1*	2001	[213,214]
HMGA2	Pi	polyhormonal	transgenic	*HMGA2* (*CMV*)	2002	[215]
p18/aSU	Pi	thyrotropinoma	multiple KO	*TP18*, *αSU*	2002	[216]
Rb^+/−^; ARF^−/−^	Pi	-	heterozygous KO	*RB1*, *ARF*	2002	[217]
Men1^ΔN/ΔN^; RIPcre(Men1^tm1.2Ctre^/Men1^tm1.2Ctre^ Tg(Ins2-cre)25Mgn/0 Tg(Ins2-cre)1Heed/0 Tg(Ins2-cre)1Dh/0)	Pi	insulinoma	TS homozygous KO	*MEN1* (*RIP*)	2003	[218]
Ink4c/p53-null	Pi	-	homozygous KO	*INK4c*, *ARF*	2003	[219]
HMGA1	Pi	polyhormonal	transgenic	*HMGA1* (*CMV*)	2005	[220]
αGSU PTTG Rb^+/−^	Pi	gonadotropinoma	transgenic	*PTTG* (*αSU*), *RB1*	2005	[221,222,223]
Cdk4^R/R^; p27^−/−^	Pi	-	homozygous KO	*CDK4* ^R/R^ *CDKN1B*	2005	[224]
Rb^+/−^; Ini1^+/−^	Pi	corticotropinoma	heterozygous KO	*RB1*, *INI1*	2006	[225]
Prkar1a^+/−^	Pi	somatotropinoma	TS heterozygous KO	*PRKAR1c*	2008	[226]
Aip^+/−^	Pi	somatotropinoma	heterozygous KO	*AIP*	2010	[227]
Tg-PCE; p27Kip1^−/−^	Pi	somatotropinoma	transgenic	*CCNE1 CDKN1B*	2010	[228]
p19Ink4d	Pi	polyhormonal	homozygous KO	*CDKN2D*	2014	[229]
Crh-1201	Pi	corticotropinoma	inducible KI	*CRH* (*mutCrh*)	2014	[230]
Rb^?/?^ Tp53^?/?^	Pi, T, Pa	MTC, ICC	multiple KO	*TP53*, *RB1*	1995	[231]
TRAMP	Pr	NEPC	transgenic	*SV40-Tag* (*PB*)	1995	[232,233,234]
CR2-Tag	Pr	NEPC, NEC	transgenic	*SV40-Tag* (*Cryptdin-2*)	1998	[235]
PSP-TGMAP	Pr	NEPC, NEC	transgenic	*SV40-Tag* (*PSP94*)	2002	[236,237]
12T-7f LPB-Tag/PB-Hepsin	Pr	NEPC	transgenic	*SV40-Tag* (*PB*)	2004	[238]
PSP-KIMAP	Pr	NEPC	KI	*SV40-Tag* (*PSP94*)	2005	[237,239]
P53^PE−/−^; Rb^PE−/−^	Pr	NEPC, adenoma	homozygous KO	*TP53*, *RB1*	2006	[240]
FG-Tag	Pr, AC	NEPC, ACT	transgenic	*SV40-Tag* (*HbF*)	1996	[241,242,243]
PTH-MEN	PT	-	TS homozygous KO	*MEN1* (*PTH*)	2003	[244]
Vil-Cre-ER^T2^ LoxP-Tag (Tg (Vil-cre) 997Gum/J)	SB, C	NEC, glandular, mixed	transgenic	*SV40-Tag* (stochastic)	2010	[245]
INS-GAS	St	NET	transgenic	*GAST* (*RIP*)	1993	[141]
bK6-HPV16e	St	NEC	transgenic	HPV-16 early region (*bk6*)	1994	[246]
Gastrin KO	St	NET	homozygous KO	*GAST*	1998	[142]
Atp4b-SV40 Tag	St	NEC	transgenic	*SV40-Tag* (*ATP4b*)	2004	[247]
Villin-Cre; Men1^loxP/loxP^	St	adenoma	TS homozygous KO	*MEN1* (*VIL1*)	2012	[248]
CEA424-SV40-Tag (Tg(CEACAM5-Tag) L5496Wzm/Cnrm)	St	dysplastic, NEC	transgenic	*SV40-Tag* (*CEA*)	2012	[249]
Atp4a^R703C/R703C^	St	dysplasia	homozygous KI	*ATP4A* ^R703C^	2016	[250]
VillinCre; Men1loxP/loxP; Sst^−/−^	St	ECL cell tumor	homozygous KO	*MEN1* (*VIL1*)	2017	[251]
CT/RET	T	MTC	transgenic	*RET*^C634R^ (*Ctct/cgrp/CGRP*)	1997	[252]
RET/PTC3	T	PTC	TS KI	*RET/Ptc3* (*TG*)	1998	[253]
RET/PTC1	T	PTC	TS KI	*RET/PTC1* (*TG*)	1999	[254]
ret/PTC1 TP53^?/?^	T	ATC	multiple KO	*RET-PTC1* (*TG*), *TP53*	2000	[255]
CALC-MEN2B-RET	T	MTC	KI	*RET*^M918T^ (*CT/CGRP*)	2000	[256]
TRK-T1	T	PTC	transgenic	*TRK-T1* (*TG*)	2000	[257]
RET/PTC3, Tp53^−/−^	T	ATC	homozygous KO	*RET-PTC3* (*TG*), *TP53*	2001	[258]
TrRbPV-PV	T	FTC	multiple KI	*TRbPV/PV* (*Tg*)	2002	[259]
Rap1b^GV12^-LoxP-N17	T	FTC	transgenic	*RAP1b*^G12V^ (*Tg*)	2004	[260]
N-RASQ61K	T	PTC	transgenic	*NRAS*^Q61K^ (*TG*)	2006	[261]
CT-RET	T	MTC	transgenic	*RET1* (*CT/CGRP*)	2010	[262]
Pten^L/L^-TPO-Cre	T	FTC	TS homozygous KO	*PTEN* (*TPO*)	2010	[263,264]
BRAF^V600E^	T	PTC	TS inducible KI	*BRAF*^V600E^ (*TG*)	2011	[265]
Pten-PPFP	T	FTC	homozygous KO	*PPFP, PTEN* (*TPO*)	2011	[266]
[Pten, p53] thyr^−/−^	T	ATC	TS homozygous KO	*TP53, PTEN*	2011	[267]
R1a-TpoKO	T	FTC	TS homozygous KO	*PRKAR1A* (*TPO*)	2012	[268]
p25OE	T	MTC	TS inducible KI	*CDK5R1* (*ENO2*)	2013	[269]
Thrb^PV/PV^; Kras^G12D^	T	ATC	homozygous KI	*THRB*^PV/PV^ (*TG*) *KRAS*^G12D^ (*TG*)	2014	[270]
BRAF^V600E^/PIK3CA^H1047F^	T	ATC	TS KI	*BRAF*^V600E^ (*TG*) *PIK3CA*^H1047R^	2014	[271]
BRAF^V600E^/PIK3CA^H1047F^	T	ATC	TS KI	*BRAF^V600E^* (*TPO*)	2014	[147,272]

^1^ ?/? indicates that multiple homo/heterozygous knockout combinations were created targeting the same genes. ^2^ Adrenal medulla (AM), anus (An), colon (C), carotid body (CB), esophagus (E), gastroesophageal junction (GEJ), jugulotympanic (JT), lung (L), pancreas (Pa), Pituitary (Pi), prostate (Pr), parathyroid (PT), rectum (R), small bowel (SB), stomach (St), thyroid (T), unknown (U). ^3^ Tissue specific (TS), knock-out (KO), knock-in (KI); multiple indicates both homozygous and heterozygous models were used. ^4^ Bovine arginine vasopressin (AVP), bovine thyroglobulin (Tg), carcinoembryonic antigen (CEA), mouse metallothionein 1 (MT), mouse insulin promoter (MIP), polyoma middle T antigen (PyMT), rat elastase promoter 1 (EL), rat glucagon promoter (RG), rat insulin promoter (RI), rat insulin-2 (RIP2), rat probasin promoter (PB), simian virus 40 large T antigen (SV40-Tag).

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
