# Peer review of "Preclinical Models of Neuroendocrine Neoplasia"

_cancers, 2022, doi:10.3390/cancers14225646_

Round 1
Reviewer 1 Report
The review article titled “Preclinical Models of Neuroendocrine Neoplasia” by Sedlack and colleagues is well-written and provides a timely overview on the current models of neuroendocrine neoplasia. The authors consider models of varying complexity and accuracy, covering both in vitro models (e.g. cell lines and organoids), and in vivo models (e.g. xenografts and GEMMs). The review provides the reader a resource and understanding of these models that will help to enable research into pathology and treatment across the spectrum of diseases.
Below are a few suggestions that may strengthen the applicability of the review for the target audience.
There is a very recent published article (Viol et al, Novel preclinical gastroenteropancreatic neuroendocrine neoplasia models demonstrate the feasibility of mutation-based targeted therapy, Cellular Oncology, 2022) that should be incorporated into this review. In this paper, the authors established and characterized 3 (GEP-NET) cell lines (NT-18P, NT-18LM, NT-36) and 2 (GEP-NEC) cell lines (NT-32 and NT-38).
All of the tables are listed numerically by year. While this provides a historical perspective (which will still be present in the text), it may be better for the reader if the items were sorted and presented by “type/source”. This presentation style would allow the reader to quickly identify the all of the cell lines/PDX/xenograft/GEMM within a particular group rather easily. For example, the average reader may want to quickly identify all of the SCLC cell lines available in the literature.
Line 123 – should be “BON1”, “BON”
While it is mentioned in the Introduction and different parts of the Results, a final summary sentence (short paragraph) should be included at the end of the Discussion highlighting the clinical importance/potential of further developing these models (i.e. drug development/screening, etc).
Author Response
Thank you for your review! Per you suggestions we have changed:
- Added the Viol et al article.
- Tables were resorted by model type.
- BON1 typo corrected.
- Summary paragraph added, including some content from the introduction that another reviewer requested be moved to the conclusion (lines 352-367).
Thank you!
Reviewer 2 Report
The work of Sedlack and colleagues summarizes current knowledge on available preclinical models of Neuroendocrine neoplasms including both in vitro models such as cell lines and 3D models, and in vivo models such as xenografts and genetically-engineered mouse models. This is a potentially interesting work, conceptually relevant and original however there are some concerns that need to be better addressed. Please find below my suggestions to improve the readability:
- I perfectly understand that given the nature of the work there are many references, however I found some of them redundant. I therefore suggest to check the bibliography by inserting only the strictly necessary references and to remove the citations of works published in journals with very low impact factor (i.e Cureus, Thoracic Surgery Clinics, J.Vis.Exp, etc.).
- In the introduction section paragraph between L87-L95 seems out of context; I suggest to move it in conclusion section.
- Section 2.1.1 seems a little bit redundant, NEN cell lines are already mentioned before in the upper section. I suggest to remove all subparagraphs in results to improve readability and to split out results section into 4 subsections: cell lines, 3D models, PDX and GEMMs.
- What authors mean with “success” and “attempts” in tables 2 and 3? It should be explained in the text or removed from tables.
- Section 2.2 title is confusing. It should be changed into 3D models instead organoids since also spheroids are mentioned
- In table 4 mouse model type should be included (transgenic, knock-in, knock-out, conditional ko, etc)
Author Response
Thank you for your helpful coments! Per you suggestions we have made the following changes:
- Redundant citations in the introduction have been removed. One citation from J. Vis. Exp. was referencing a model that is not yet published elsewhere and has been retained.
- Moving editorializing statements into the conclusion.
- We were not sure what you meant here since the results are already split into subsections.
- This has been clarified in table captions.
- The terminology on organoids/3D models has been fixed throughout.
- Mouse model genetic types have been added.
Thank you!
Reviewer 3 Report
This is a comprehensive review of the preclinical models of neuroendocrine neoplasia. Given the heterogeneity of the disease it provided a thorough review of the various efforts in the main subtypes of NEN and NEC, but also by primary site of disease. This effort highlights as other similar reviews have done in the past, that the nature of this disease limits the traditional preclinical models utilized to characterize other cancer.
As a review it kept to the objectives to discuss historical context and current use and limitations. I agree that NEC should be included as this is a review of all NENs, but due to the notable differences between the two, review of NETs alone may have permitted a deeper assessment of the limitations seen with NETs where much of the work is needed.
Author Response
Thank you so much for your feedback!